# Hijacking Robot Teams Through Adversarial Communication

**Zixuan Wu**      **Sean Ye**      **Byeolyi Han**      **Matthew Gombolay**
Georgia Institute of Technology, Atlanta, GA, USA
{zwu380, seancye, bhan67, mgombolay3}@gatech.edu

**Abstract:**

Communication is often necessary for robot teams to collaborate and complete a decentralized task. Multi-agent reinforcement learning (MARL) systems allow agents to learn how to collaborate and communicate to complete a task. These domains are ubiquitous and include safety-critical domains such as wildfire fighting, traffic control, or search and rescue missions. However, critical vulnerabilities may arise in communication systems as jamming the signals can interrupt the robot team. This work presents a framework for applying black-box adversarial attacks to learned MARL policies by manipulating only the communication signals between agents. Our system only requires observations of MARL policies after training is complete, as this is more realistic than attacking the training process. To this end, we imitate a learned policy of the targeted agents without direct interaction with the environment or ground truth rewards. Instead, we infer the rewards by only observing the behavior of the targeted agents. Our framework reduces reward by 201% compared to an equivalent baseline method and also shows favorable results when deployed in real swarm robots. Our novel attack methodology within MARL systems contributes to the field by enhancing our understanding on the reliability of multi-agent systems.

**Keywords:** Adversarial Attacks, Multi-Agent Reinforcement Learning

## 1 Introduction

Effective communication among robots is essential for information exchange, collaboration, and collective decision-making. It plays a vital role in various robotics domains, including collaborative manipulation [1] and multi-robot navigation [2]. Ensuring reliable and secure communication is crucial for maintaining the overall system's performance, safety, and integrity.

Multi-agent reinforcement learning (MARL) has been a powerful tool to train agents in complex domains but the literature is lacking in studies about the vulnerabilities and defenses of these systems. MARL techniques that utilize communication have been widely applied to scenarios where multiple agents need to collaborate for a shared goal in robotics tasks such as autonomous driving [3, 4, 5] and path planning [2, 6, 7]. Researchers examined various aspects of communication in MARL, including when to communicate [8], who to communicate with [9], and different types of graph-structured communication [10, 11]. Some of these frameworks use binarized communication to pursue Low-Size, -Weight, and -Power (Low-SWAP) systems [11] which causes agents to communicate in a highly efficient manner. Malicious actors can compromise these systems and endanger the lives of many [12].

There are only a few works that assume the communication channel can be imperfect [13] or can be attacked [14] paralleling with its computer vision counterpart [15, 16, 17, 18, 19], which makes it crucial to understand these weaknesses such that we can create appropriate defense mechanisms. In this work, we learn to attack the communication signals within a multi-robot team discretely

7th Conference on Robot Learning (CoRL 2023), Atlanta, USA.

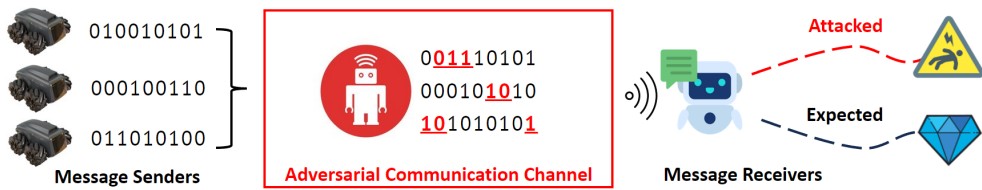

Figure 1: Adversarial Communication Pipeline: Multi-agent team (left) communicates information for decentralized coordination; an adversarial system (middle) learns a model of the teams' activities and communication patterns and (right) broadcasts counterfeit team messages to trick team members towards pursuing low-priority activities.

without any trace of the training process on the target robots. First, we learn surrogate policies from the observation, messages, and actions of the target robots which are accessible from malware or insecure networks [20, 21, 22, 23, 24]. Second, we estimate the agent rewards from their behaviors instead of using rewards from the environment. Finally, we use an actor-critic framework completely offline to learn how to hijack the targeted system without environment interactions. Our method requires the least prior knowledge as compared to prior work and results in robots traversing to the wrong location and drastically hindering team performance.

**Contributions:**

1. We propose an actor-critic framework that enables our adversarial policy to learn without direct interaction with the target agents or the environment nor the ground truth agent reward. Our framework manipulates the behaviors of target robots with the communication attacking strategy learned through surrogate target policies and transferable to real ones. Additionally, we introduce a differentiable framework for training adversarial communication policies that can modify digital communication signals [10, 11].

2. We demonstrate the effectiveness of our algorithm in three distinct multi-agent domains: predator-capture-prey, partially observable predator-prey, and speaker-listener. Across these domains, our method surpasses the baseline approach by reducing the reward of the target agents by 465% compared to a baseline approach.

3. We validate the applicability of our algorithm on physical swarm robots in the Robotarium [25]. By acting as a strong adversary, our method reduces the reward achieved by the target agents by an average of 201% across all three environments compared to a baseline strategy, which employs an equivalent random flipping approach.

## 2 Related Work

In this section, we describe how multi-agent reinforcement learning can be used to control robots in a Dec-POMDP setting. We also provide an overview of the role of communication in a MARL framework and highlight the vulnerability of communication to adversarial attacks, leading to potential system failures and safety risks.

### 2.1 Multi-Agent Reinforcement Learning

In recent years, communication has played a crucial role in enhancing coordination and collaboration among robots in multi-agent reinforcement learning (MARL) frameworks [8, 9, 10, 26, 11, 27, 28]. Compared to previous works in learning policies [29, 30], recent MARL frameworks have enabled learning in more complex environments and train multiple agents. These MARL communication frameworks have been used in several robotics applications such as multi-robot path planning [2] and cooperative driving [31]. Various approaches have been proposed, including trainable differentiable communication channels [26, 27], partially observable environments [28], and soft-attention networks for selective communication [8, 9]. However, the effectiveness of communication is threatened by adversarial attacks, which can lead to system failures and safety risks, particularly in domains like self-driving vehicles [32]. This paper aims to evaluate the robustness of communication

in MARL systems, building upon prior advancements in communication techniques of the binarized communication approach [11] that improves bandwidth efficiency. Key to note is that our adversarial policy does not make any assumptions on targets other than inter-agent binarized communication. Our adversarial attack can hypothetically be utilized in any framework as long as we can learn good surrogate policies of target agents. We aim to show the generality of our approach in future work.

## 2.2 Adversarial Attacks in MARL and Communication

Adversarial attacks were first studied within the context of computer vision, where small perturbations to the input could induce faulty outputs [18]. Adversarial attacks aim to deteriorate model performance in tasks like classification [16, 17, 19], segmentation [33, 34, 35], or object detection [36, 37]. These ideas were later extended to reinforcement learning [38, 39], altering agent actions through perturbations in environment observations [40].

In adversarial attacks, two common categories are white-box attacks, which assume knowledge of neural network weights, and black-box attacks, which assume limited model parameter information. White-box attacks typically optimize objectives using methods such as Fast Gradient Sign Method (FGSM) [16, 17] or Projected Gradient Descent (PGD) [19]. Meanwhile, black-box attacks rely on surrogate models that approximate decision boundaries such that the adversarial attacking targeting on it could be transferred to the original models. Input-output pairs in black-box scenarios can also be augmented by FGSM [18] or PGD [19] to generate a synthetic dataset that induces similar decision boundaries. We utilize black box attacks with augmentation similar to FGSM for first-order approximation of our surrogate models.

Prior work [41] has trained an adversarial communication protocol in a multi-agent setting by using a reinforcement learning agent to optimize the adversarial policy. However, this approach requires direct interaction with the environment and is impractical as the training process would be easily detected by observers of the system. To address this limitation, we propose a black box setting and employ transfer attacks [42], training a surrogate model to mimic the target model. Furthermore, our work distinguishes itself by assuming a "man-in-the-middle" attack, where an interceptor subtly flips the aggregated binarized communication vectors. This is in contrast to previous assumptions of a single target victim agent with a limited number ($\leq \frac{N-1}{2}$) of potentially malicious messages [14].

## 3 Problem Formulation

We ground our problem in a Decentralized Partially Observable Markov Decision Process (Dec-POMDP) formalism which is a 10-tuple of $\langle S, M, A, P, R, \Upsilon, O, \Pi, N, \gamma \rangle$. $S$ is the state set of the environment and $M$ is the message state set. For each agent $i \in N := 1, ..., N$, the agent chooses an action $a_i \in A$ at state $s_i \in S$. The transition function is denoted by $P(S'|S, A)$. Each agent has its own reward based on global state-actions $r_i = R_i(S, A)$ and $\gamma \in [0, 1)$ as the discount factor. Since the environment is partially observable, each agent also has its own individual partial observation $v_i \in \Upsilon$ which is produced by the observation function $\Upsilon = O(S)$. Agents can have two policies: an action policy $\pi_i^a(a_i|\tau_i)$ and a message policy $\pi_i^c(m_i^{out}|\tau_i)$. These are both conditioned on their own partial observations and the messages received from other communicative agents $\tau_i = \{v_i, M \backslash m_i\}$.

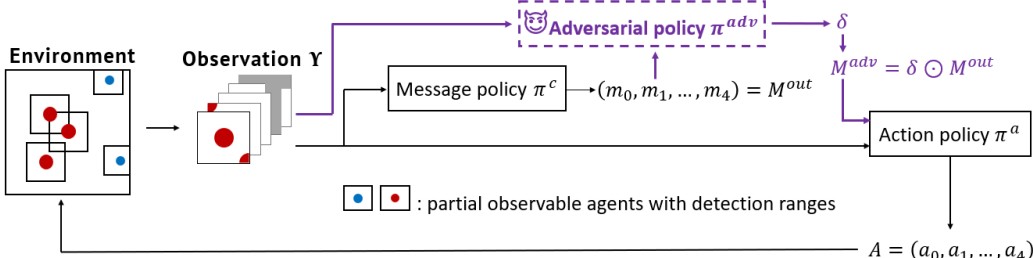

Figure 2: Adversarial Policy at Test-Time: Our adversarial attacking policy $\pi^{adv}$ changes the messages from the message policy $\pi^c$ such that the receiving agents are maximally disrupted.

We model a single adversarial policy $\pi^{adv}(\delta|\Upsilon, M^{out})$ that has access to the partial observations $\Upsilon$, message outputs $M^{out}$ of all agents and produces adversarial signal $\delta$. The partial observations could be gathered by the adversary by shadowing the relevant targeted agents. We generate a malicious revised communication, denoted as $M^{adv}$, by combining the perturbation $\delta$ with the original messages $M^{out}$ sent to each agent, formulated as $M^{adv} = \delta \odot M^{out}$. Figure 2 shows how the adversarial policy $\pi^{adv}$ influences the target policies. Here we consider the transition and policies as deterministic ones: $S' = P(S, A)$, $a_i = \pi_i^a(\tau_i)$, $m_i^{out} = \pi_i^c(\tau_i)$ and $\delta = \pi^{adv}(\Upsilon, M^{out})$.

## 4 Methodology

In this section, we define the design choices for training our adversarial communication policy. The goal of the policy is to minimize the reward of the victim agents while minimizing the difference between the original and tampered communication vector. Algorithm 1 provides an overview of our overall training procedure: 1) Learning a surrogate policy, 2) Learning an actor-critic for the adversarial policy, and 3) Updating the actor with differentiable binary communication. We presume access to the observations, messages, and actions of the target agents, assuming that we have successfully intercepted the communication protocol of the multi-agent system. However, we do not assume access to ground truth rewards. We also assume a binary communication channel of 16-bits but further studies could extend our work to remove this assumption.

---

**Algorithm 1:** Adversarial Communication Pseudocode

---

Input: $\mathcal{D}(\Upsilon(\text{Observations}), M(\text{Messages}), A(\text{Actions}))$
**for** $i = 0, 1, 2 \dots$ **do**
    **for** *agent j=1 to N* **do**
        Sample batch of observation, message, actions: $(v_j, v_j') \sim \Upsilon, (m_j, m_j') \sim M, a_j \sim A$
        Update the surrogate policy $\pi_j^{surr}(a_j|v_j, m_j)$
    **end**
    Compute targets

$$R^{adv} = -\frac{1}{N}\sum_j log(\pi_j^{surr}(a_j|v_j, m_j)) \tag{1}$$

$$y = R^{adv} + \gamma Q_{\phi, trgt}(\Upsilon', A')|_{A' = \Pi_{trgt, surr}^a(\tau_i')} \tag{2}$$

    Update the adversarial critic with one step of gradient descent

$$\phi \leftarrow \phi - \nabla_\phi(Q_\phi(\Upsilon, A) - y)^2) \tag{3}$$

    Update the adversarial policy with one step of gradient descent

$$\theta \leftarrow \theta + \nabla_\theta(Q_\phi(\Upsilon, A^{adv}) - \mathcal{C}_{flip})|_{A^{adv} = \Pi_{surr}^a(v_j, m_j^{adv}), M^{adv} = \delta \odot M^{out}, \delta = \pi_\theta^{adv}(\Upsilon, M^{out})} \tag{4}$$

**end**

---

First, we train surrogate policies to imitate the real policies (Section 4.1). We can then learn a Q-function for the adversarial policy by assigning rewards based upon the surrogate policies (Eq. 1) and using the Bellman equation to update the critic (Eq. 2, 3). Details are described in Section 4.3. Finally, the adversarial policy is updated with the differentiable binary flipping mechanism (Eq. 4) and described in Section 4.2. We include hyperparameters in Appendix B.2 for more details.

### 4.1 Learning a Surrogate Policy

As our method is a black-box attack, we assume we do not have access to the ground truth policies of the agents we are attacking. To this end, we learn a *surrogate policy ($\pi^{surr}$)* for each agent we are attacking, where we assume access to a dataset $\mathcal{D}(\Upsilon, M, A)$ consisting of observations,

communication, action pairs from victim agents. The surrogate policies are used in two ways as described in the next section: 1) as a reward signal to learn a critic and 2) as a mechanism to simulate the adversarial output into real agent actions. To obtain a similar first-order approximation of our surrogate policies to the real policies, we augment our dataset with neighboring data by adding small Gaussian noise to each input to produce augmented input-output pairs. We find that this augmentation is enough to obtain a useful surrogate policy and thus there is no need to achieve higher precision using much more complicated methods like FSGM augmenting [18]. We use behavioral cloning methods to minimize the log-likelihood between actions given the agent observations and messages from other agents ($\log p(a|v, m)$).

## 4.2 Differentiable Targeting of Binary Communication Channels

In binary communication, each bit has two states: 0 and 1. Modifying a bit involves flipping it to the other state. To make this process differentiable, we need to parameterize it similarly to its continuous counterpart. We define the adversarial modified message $M^{adv}$ as the composition of the original communication vector $M^{out}$ and the parameterized modification $\delta$ [41].

Two methods are available to parameterize the adversarial policy. One method is to directly output the adversarially revised communication vector as $M^{adv} = \delta = \pi^{adv}(M^{out}, v)$, which we call the "direct" form. The second method is called "flipping" whose adversarial policy output $\delta$ is used to indicate which digit to flip such that we can write it in an XOR form using boolean algebra as in Equation 5 where $\cdot$ means pointwise multiplication.

$$M^{adv} = \delta \cdot \overline{M^{out}} + \overline{\delta} \cdot M^{out} = \delta \cdot (1 - M^{out}) + (1 - \delta) \cdot M^{out} \tag{5}$$

## 4.3 Learning an Actor-Critic for Adversarial Communication

Our goal for the adversarial policy is to **secretly** train itself without any interaction with the environment in a black-box setting such that the adversarial training process does not induce any abnormalities and cannot be detected. This requirement raises a higher standard than recent work [41], where the adversarial policy is trained with reinforcement learning and requires environment interactions of the adversarial policy's actions. We adapt the actor-critic framework to learn 1) a critic ($Q_\phi(\Upsilon, A)$) that is learned within the observation-action ($\Upsilon, A$) space of the target agents and 2) an actor policy that distorts the communication messages. We use the actor-critic framework as it allows us to utilize the original Dec-POMDP of the target agents for the critic rather than building a new separate MDP within the space of the observations and messages as actions. We train our Q-function $Q_\phi(\Upsilon, A)$ to use the observation-actions of all agents in the environment based on the Bellman equation and TD error as shown in Equation 6.

$$\mathcal{L}(\phi) = \mathbf{E}_{\Upsilon, A, R^{adv}, \Upsilon'}[(Q_\phi(\Upsilon, A) - y)^2], \ y = R^{adv} + \gamma Q_{\phi, targ}(\Upsilon', A')|_{A' = \Pi^a_{trgt, surr}(\tau'_i)}. \tag{6}$$

A question naturally arises: how do we get the reward $R^{adv}$ to train this Q-function? Because we do not assume access to ground-truth rewards as previous literature does, we cannot utilize the negative mean of all agent reward $R^{adv} = -\frac{1}{N} \sum_{i=1}^{N} r_i$ to optimally degrade performance on their own metrics. Instead, we assign the reward for a certain state-action pair of all agents to be the inverse of the log probability of the optimal action from the surrogate policy (Eq. 7).

$$R^{adv} = -\frac{1}{N} \sum_j log(\pi_j^{surr}(a_j | v_j, m_j)) \tag{7}$$

Intuitively, we are driving the critic to punish state-action pairs visited by the target policies. Another interpretation is that the probability of the samples' appearance is proportional to the exponential of the reward in inverse reinforcement learning (IRL) theory [43, 44].

Given a well-trained critic, we can learn the adversarial policy $\pi^{adv}(\delta | \Upsilon, M^{out})$ to modify the communication messages to maximize the Q-function minus a bit flipping penalty $\mathcal{C}_{flip} = \mathbf{L}_1(\delta)$. We utilize the surrogate policies again to produce hypothetical actions given the modified communication vectors.

$$\delta = \pi^{adv}(\Upsilon, M^{out}), \ M^{adv} = \delta \cdot (1 - M^{out}) + (1 - \delta) \cdot M^{out}, \ A^{adv} = \Pi^a_{surr}(v_j, m_j^{adv}) \tag{8}$$

The adversarial policy can then be updated through automatic differentiation to produce messages that disturb the surrogate policies. With our method, we can train the adversarial communication policy completely offline with the only assumption being that we have intercepted some observation, message, and action pairs from the target policies.

## 5 Results and Discussion

### 5.1 Environments

We utilize three domains originally proposed by the MADDPG [28] paper and modified for our use: Predator-Capture-Prey, Partially Observable Predator-Prey, and Speaker-Listener. In all environments, the goal is to maximize the number of collisions between the agents and the target. Further details of these environments are included in Appendix B.1.

**Predator-Capture-Prey (PCP)** In this environment, a team of agents must capture an adversary prey opponent. To emphasize the role of communication in this domain, capture agents cannot see any other agents and have to make decisions based on the received messages from all observing agents.

**Partially Observable Predator-Prey (PO-PP)** We modify the predator-prey environment such that all predator agents can only receive the location of the prey when they are within a distance $d$ of the prey. The agents must communicate with each other to locate the position of to the prey. We remove the capture agents from this environment.

**Speaker-Listener (SL)** In speaker-listener, a team of two agents, consisting of a *speaker* and a *listener* must work together for the listener to reach a target color destination. The speaker must communicate the target color to the listener and the listener must then go to the color destination.

### 5.2 Adversarial Communication Validation

In this section, we validate our secret adversarial communication channel performance by comparing it with a *random flipping* method where we flip the same number of bits as the adversarial communication. We ensure that these two methods are always flipping the same number of bits in the communication and compare the results across various numbers of bits flipped. The random flipping baseline has been used in prior work [14] and represents a non-adaptive black-box attacker. Therefore, we modify the adversarial policy loss function to $\mathcal{L} = -Q(v, a, m) + \alpha \cdot 1/N_c \cdot \mathbf{L}_1(\delta)$, which regularizes the policy loss by the average sum of bits flipped. The coefficient $\alpha$ is used to balance the adversarial policy performance with the number of bits flipped and we control the regularization speed by annealing $\alpha = \alpha_0 \cdot \max(n_e - \beta, 0)/\epsilon$ (where $n_e$ is the training episode number, $\beta$ is the regularizer intercept, and $\epsilon$ is the regularizer slope) to fine-tune the bits flipped. We evaluate the reward and collision statistics by running 50 episodes for each adversarial policy checkpoint. Training curves with the regularized loss and hyperparameteres can be found in Appendix B.2.

Figure 3 shows the rewards and collisions versus the episodes from which we get our adversarial policy in the three environments. Agents' reward and collisions from the adversarial communication increase as the number of bits flipped decreases but are always less than those from random flipping. This result validates the adversary property of our method and distinguishes it from random noise with the same magnitude. We also find that the gap between the two methods is lower when the number of bits flipped is lower, which is reasonable since it is difficult for adversarial policy to attack the critical combinations of the communication digits in such a limitation. Interestingly, the random flipping curve nearly remains horizontal in the PO-PP environment, which means that random attack does not work at all regardless of the number of bits flipped. This is because the predators do not only rely on communication but also on their own observations to take actions, therefore, irregularly changing the communication will not confuse agents much, compared to the adversarial policy which flips crucial bits and guides the agents to low-reward regions.

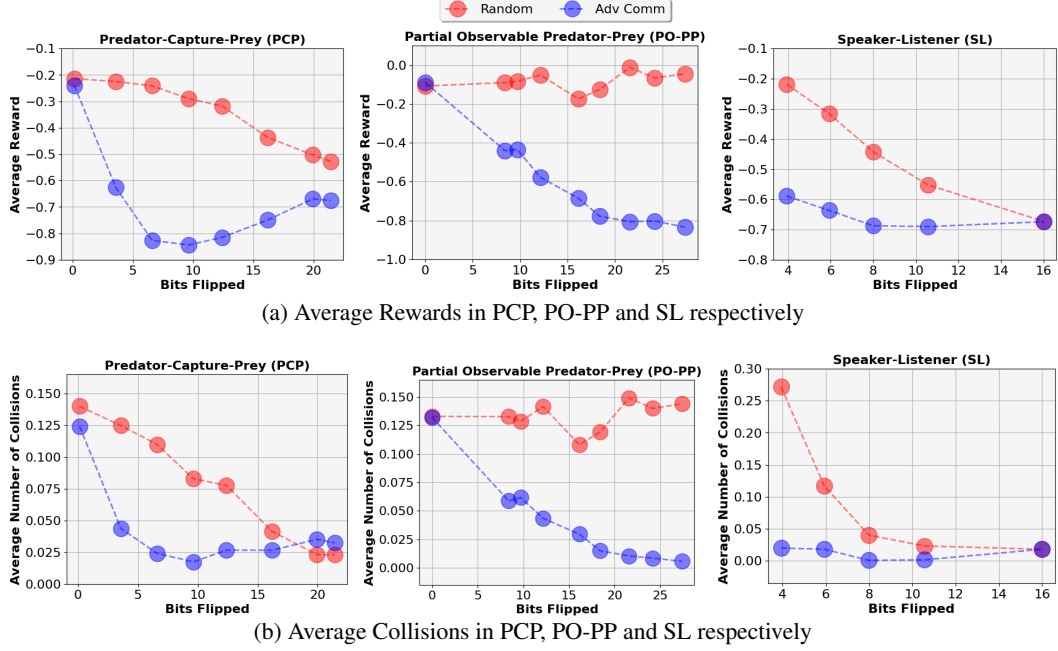

(a) Average Rewards in PCP, PO-PP and SL respectively

(b) Average Collisions in PCP, PO-PP and SL respectively

Figure 3: The figure illustrates the comparison between the adversarial policy (blue) and random flipping (red) in terms of average reward and number of collisions. Consistently, the adversarial communication policy degrades team performance more effectively than the random policy across all bit flip counts, leading to lower rewards and a lower number of collisions for agents.

## 5.3 Comparison of Adversarial Message Parameterization

In this section, we compare our adversarial policy, between the "flipping" regularization strategy (as described in Section 4.2), and the "direct" strategy in terms of the normalized adversarial policy performance score ($Sc$) which denotes how much worse an agent performs per flipping a single digit (Appendix B.3). The training and testing settings are the same as section 5.2 and we record the maximum normalized score of the flipping method (ours) to the direct method (Table 1).

Table 1: Reward and Collision Normalized Scores

|  | PCP | | PO-PP | | SL | |
| --- | --- | --- | --- | --- | --- | --- |
|  | Reward Sc | Collision Sc | Reward Sc | Collision Sc | Reward Sc | Collision Sc |
| **Adv[Ours]** | **0.11±0.05** | **4.58±2.23** | **0.04±0.02** | **1.45±0.55** | **0.13±0.06** | **31.38±6.51** |
| **Adv[Direct]** | 0.01±0.01 | 1.07±0.14 | 0.02±0.01 | 0.80±0.07 | 0.11±0.05 | 30.72±6.19 |
| **Random** | 0.01±0.01 | 0.96±0.29 | 0.00±0.01 | 0.30±0.88 | 0.04±0.02 | 13.68±3.19 |

In the PCP and PO-PP environments, the direct strategy exhibits significantly worse reward and collision scores, performing at 90.9% and 76.64% lower, respectively, compared to our approach. This discrepancy arises from the failure of the direct strategy to effectively balance lowering agent performance and reducing flipped bits. In these environments, the direct strategy results in 38.7% and 65% bits flipped, whereas our approach achieves 7.5% and 28.3% of bits flipped. Interestingly, in the SL environment, where there is a single message sender, one receiver, and 16 bits of communication, the direct strategy performs relatively better due to the simpler balancing of performance and regularization terms. Nevertheless, our approach still outperforms the direct strategy by 18.18% and 2.15% in terms of reward and collision, with 8.15% and 8.29% of bits flipped, respectively. These findings highlight the critical importance of the flipping representation in facilitating backward gradient computation.

## 5.4 Robotarium Physical Robot Demonstration

We demonstrate our results on a physical swarm robotics system (details in Appendix A). We utilize state-based position control to drive each robot according to their policies both with the adversarial

communication intervention and random flipping intervention. We show that our adversarial communication policy drastically reduces reward in all three environment settings (Table 2) with the same number of bits flipped per episode and averaged over three episodes.

Table 2: Attacked Agents Reward and Collision

|  | PCP | | PO-PP | | SL | |
| --- | --- | --- | --- | --- | --- | --- |
|  | **Reward** | **Collision** | **Reward** | **Collision** | **Reward** | **Collision** |
| **Adv[Ours]** | **-0.71±0.34** | **0.05±0.23** | **-0.71±0.34** | **0.05±0.23** | **-0.43±0.14** | **0.00±0.00** |
| **Random** | -0.18±0.33 | 0.29±0.45 | 0.02±0.56 | 0.20±0.40 | -0.21±0.15 | 0.34±0.47 |

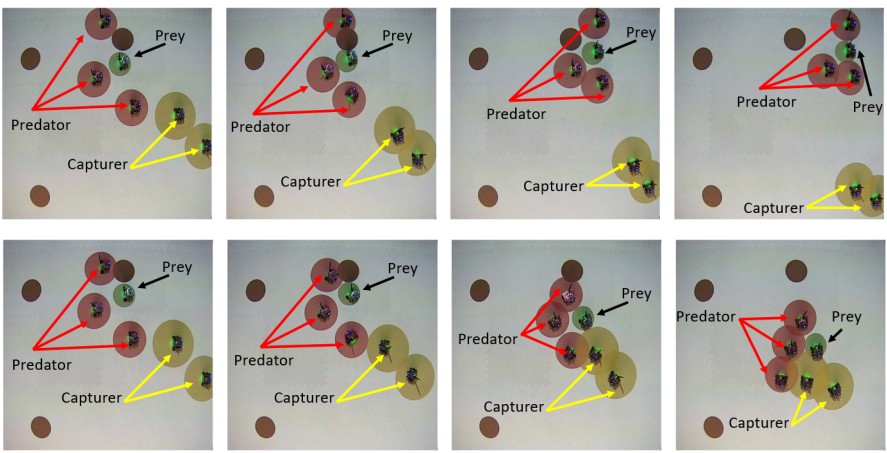

Figure 4: PCP Demonstration: Adversarial Communication (top) vs Random Flipping (bottom)

# 6 Limitations and Future Work

Our work relies on the assumption that the surrogate policies can accurately model the behaviors of the true policies and depends on using the agent policies to estimate the ground-truth reward function. A limitation of this approach is the amount of training data required to imitate the surrogate policy, where the target agents may detect that information is being collected. Additionally, IRL methods [44] could be used to learn a reward function that better reflects the ground truth reward.

Our work has important ethical implications as it could be used to attack important systems but can improve the community's ability to help improve the robustness of these systems by characterizing the vulnerabilities. This work does not assume any strategies for the target agents to defend against adversarial attacks. A defender may include a parity bit indicating whether the total number of 1-bits is even or odd to defend against our attacks on the communication. Additionally, defenders could learn a response strategy by adjusting its communication scheme if they had access to previous experiences with the attackers. Finally, if the adversarial attack is conducted on a multi-agent system with interpretable communication vectors, it may be easy to identify that messages have been altered. We leave adversarial attacks in this setting for future work.

# 7 Conclusion

We introduce a practical adversarial communication policy that does not need direct environment interaction, enhancing the feasibility of adversarial attacks. Our method also utilizes a robust approach for estimating agent rewards from observing behaviors only, without reward information. Lastly, we pioneer a differentiable method for adversarial communication in discrete binary channels, flipping bits for improved attack efficacy. Our algorithm is validated on real swarm robots in the Robotarium platform. This showcases the versatility and real-world applicability of our approach. Our framework opens new avenues for enhanced security and robustness in multi-agent systems, with potential implications across various domains.

**Acknowledgments**

We wish to thank our reviewers for their valuable feedback in revising our manuscript. We also thank Letian Chen for his expertise in inverse reinforcement learning, which inspired many of our ideas. Additionally, both Rohan Paleja and Esi Seraj provided invaluable insights on multi-agent reinforcement learning. This work was sponsored by the Naval Research Laboratory under grant number N00173-21-1-G009.

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

# Appendix A   Real-World Demonstrations: Robotarium

We use Robotarium [20], a free remotely accessible swarm robotics research platform, to do real-world demonstrations. It is equipped with a group of miniature differential drive robots 'GRITSBots' on a testbed measuring 130×90×180 cm, with a projector and an automatic overhead tracking system. The GRITSBot's main board has WiFienabled 160 MHz ESP8266 chip as the controller and communication (54 MBit/s WiFi) and the stepper motors droven by Atmega 168 microcontroller [20]. The global position is tracked using an overhead camera and then used down-stream for safety checking and feedback control. Features of the robot environment are displayed by the projector for visualization.

First, we need to run our algorithm in the robotarium simulator before implementing it on the real platform. However, physical collisions are strictly prohibited when using actual robots. To overcome this limitation, we record the trajectories of each agent in the real environment and perform post-analysis to determine if there are any instances where two robots collide. This analysis is based on the relative distance between the robots, following our predefined criteria. A collision between two robots is defined as when the circles centered on each robot intersect. The radius of each circle is defined according to the environment specifications [23]. The reward for the environments is defined as the L2 distance between the robot and its target destination. In PP and PCP, the target is the prey robot. In SL, the target is the designated goal location. We show the trajectories we collected in each environment Fig 5.

We also include the average reward and collision numbers of each robot in Tables 3-8. It shows that our adversarial method universally outperforms the random flipping one for each agent since it makes the attacked agents receive less reward and has fewer collisions with their targets. Moreover, we find that our method is even more stable than the random flipping one, with standard deviation only decreasing by 22.97%, 53.33%, and 40.89% on average in the three environments.

Table 3: PCP Reward

|  | Capture Agent 1 | Capture Agent 2 | Average |
|---|---|---|---|
| **Adv[Ours]** | **-0.76±0.32** | **-0.66±0.36** | **-0.71±0.34** |
| **Random** | -0.27±0.29 | -0.090±0.36 | -0.18±0.33 |

Table 5: SL Reward

|  | Listener |
|---|---|
| **Adv[Ours]** | **-0.43±0.14** |
| **Random** | -0.21±0.15 |

Table 4: PCP Collisions

|  | Capture Agent 1 | Capture Agent 2 | Average |
|---|---|---|---|
| **Adv[Ours]** | **0.02±0.14** | **0.09±0.28** | **0.05±0.23** |
| **Random** | 0.17±0.37 | 0.40±0.49 | 0.29±0.45 |

Table 6: SL Collisions

|  | Listener |
|---|---|
| **Adv[Ours]** | **0.00±0.00** |
| **Random** | 0.34±0.47 |

Table 7: PO-PP Reward

|  | PO Agent 1 | PO Agent 2 | PO Agent 3 | PO Agent 4 | Average |
|---|---|---|---|---|---|
| **Adv[Ours]** | **-0.81±0.34** | **-0.80±0.27** | **-0.71±0.31** | **-0.80±0.30** | **-0.71±0.34** |
| **Random** | 0.01±0.56 | -0.06±0.54 | -0.00±0.60 | -0.03±0.54 | 0.02±0.56 |

Table 8: PO-PP Collisions

|  | PO Agent 1 | PO Agent 2 | PO Agent 3 | PO Agent 4 | Average |
|---|---|---|---|---|---|
| **Adv[Ours]** | **0.00±0.06** | **0.00±0.00** | **0.00±0.00** | **0.00 ±0.05** | **0.05±0.23** |
| **Random** | 0.26±0.44 | 0.06±0.33 | 0.33±0.47 | 0.15±0.35 | 0.20±0.40 |

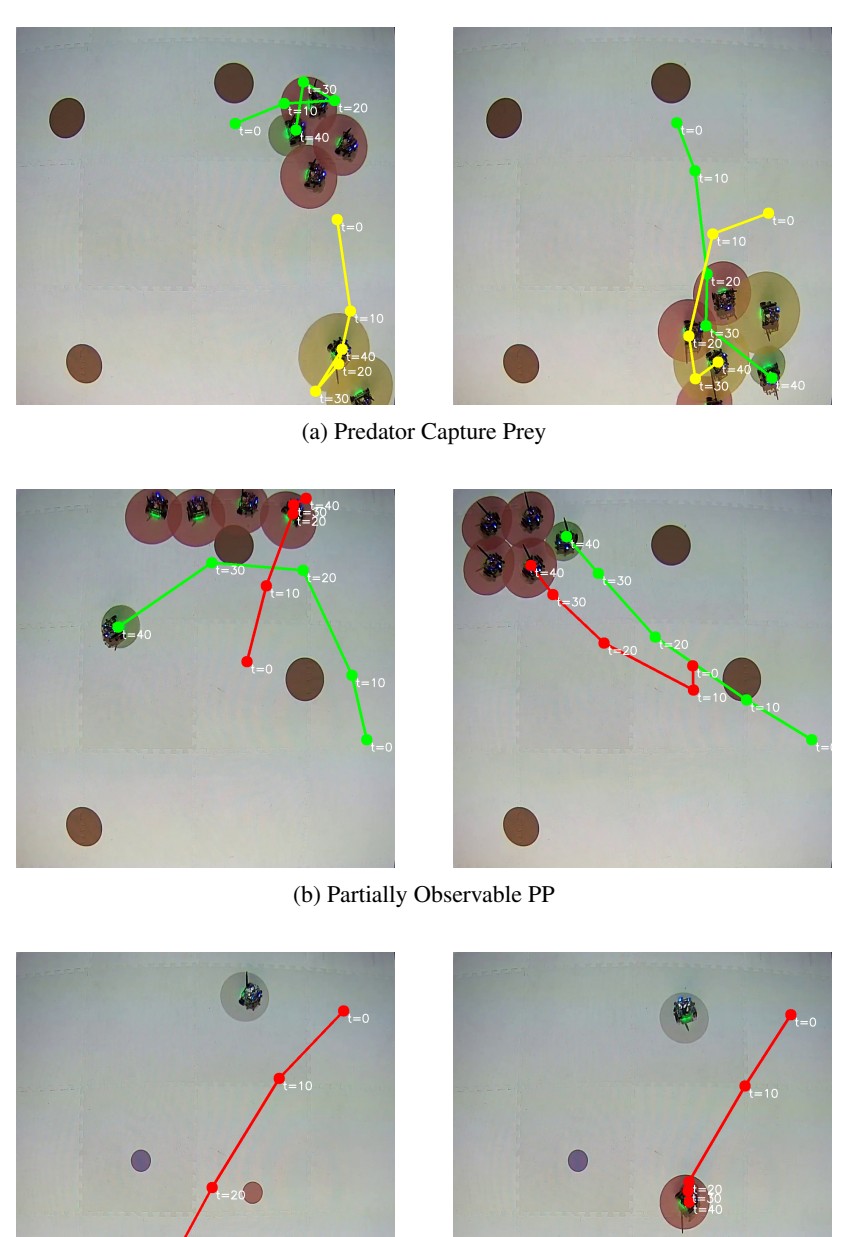

(a) Predator Capture Prey

(b) Partially Observable PP

(c) Speaker Listener

Figure 5: Comparison of Environment Trajectories: All three environments are shown, where the left images are the adversarial communication policy rollouts and the right images are the random flipping rollouts.

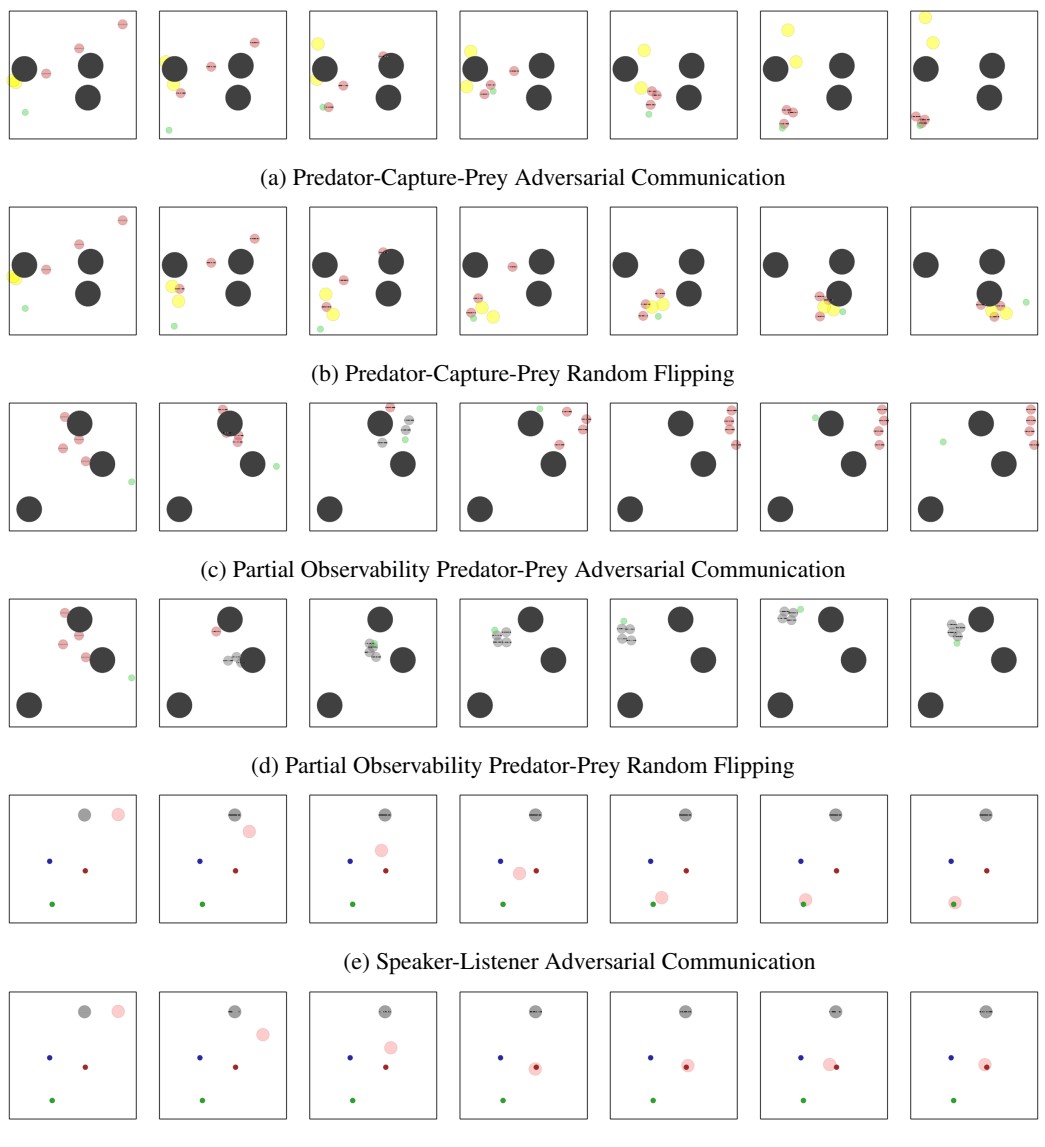

Figure 6: These image series show the performance of agents when applying our adversarial communication and random flipping strategy in three environment: Predator-Capture-Prey (a, b), Partial Observability Predator-Prey (c, d) and Speaker-Listener (e, f).

# Appendix B Simulation Experiment Details

## Appendix B.1 Domains

Here we show the hyperparameters used in each environment training (Table 9) and qualitative results (Figure 6).

In the PCP environment (a, b), the predators (also called perception agents) are shown as red which can observe all other agents, however, the yellow capture agent (also called action agents) are blind and can only know where the prey (green) is by receiving the messages from the predators. Therefore, communication is the only useful information based on which the capturers can make decisions. Each capture agent will receive a 16-bit communication from each of the three predators so we intercept 48 bits and modify them with our adversarial policy. Compare with Figure 6(a) and 6(b), we find that our adversarial policy can successfully push the captures agents away from the prey but the random flipping one cannot stop the capture agents from pursuing the prey with the same number of bits flipped.

We observe similar behaviors in PO-PP when we compare Figure 6(c) and 6(d), in which the predators and prey are shown with red and green. The difference between PO-PP and PCP environments is that we remove the capture agents but change the predators to be partially observable agents which can only see the prey within a certain distance. Predators change color from red to grey if they observe the prey. If one predator observes the prey, it can broadcast this information to others with its 16-bit communication so that the team can cooperate with each other to achieve higher rewards. When we apply the adversarial policy (see Figure 6), we find that the predators just ignore the prey even though they see it and never collaborate to collide with the prey compared with the random flipping one in Figure 6.

In the speaker-listener environment (Figure 6(e, f)), the speaker knows the colored goal the listener should go to but the listener does not. However, the listener knows the position of the three colored goals. Therefore, the speaker needs to learn to communicate the correct color within its 16-bit communication and the listener should learn which color it needs to go to from the message. Our adversarial method (6) can make the listener go to a completely wrong colored destination, while the random flipping method cannot because it cannot attack the crucial bits of the communication.

## Appendix B.2 Training Details

We show the hyperparameters for all environments in Table 9.

Table 9: Hyperparameters for training and testing PCP, PO-PP and SL

| Hyperparameter | Environment | Value |
|---|---|---|
| Buffer Length | PCP, PO-PP, SL | 1048576 |
| Episode Number | PCP, PO-PP, SL | 50001 |
| Episode Length | PCP, PO-PP, SL | 100 |
| Batch Size | PCP, PO-PP, SL | 1024 |
| Discount Factor $\gamma$ | PCP, PO-PP, SL | 0.9 |
| Learning Rate | PCP, PO-PP, SL | 0.0001 |
| Regularizer Coefficient $\alpha_0$ | PCP, SL | 0.1 |
| Regularizer Coefficient $\alpha_0$ | PO-PP | 0.004 |
| Regularizer Intercept $\beta$ | PCP, PO-PP, SL | 3000 |
| Regularizer Slope $\epsilon$ | PCP, PO-PP, SL | 20000 |
| Perception Threshold $\eta$ | PCP, PO-PP, SL | 3 |

In Figure 7, we show the reward and collision numbers over the training procedure. We start the training without the bit-flipping regularizer term $\mathcal{C}_{flip}$. Then, at episode $\beta$ (3000), we begin to regularize the adversarial policy to flip fewer and fewer bits. As training procedes, the regularization term

dominates the training process and the agents rewards increase as fewer bits are flipped. However, we see that our adversarial agent policy outperforms the random policy at every training iteration.

### Appendix B.3   Normalized Score

The normalized score is defined as:

$$S = \frac{RC_{no\_adv} - RC_{adv}}{\max(N_f, \eta)} \tag{9}$$

where $RC_{adv}$ and $RC_{no\_adv}$ represent the reward or the collision number with and without applying the adversarial policy and their difference represents how much the adversarial communication channel degrades the agent performance. It is then normalized by the the number of bits flipped with a perception threshold $\eta$ which increases the numeric stability in case of extremely small flipping number $N_f$. A higher score signifies that, for each bit flipped, the adversarial approach has a greater detrimental impact on the team's performance.

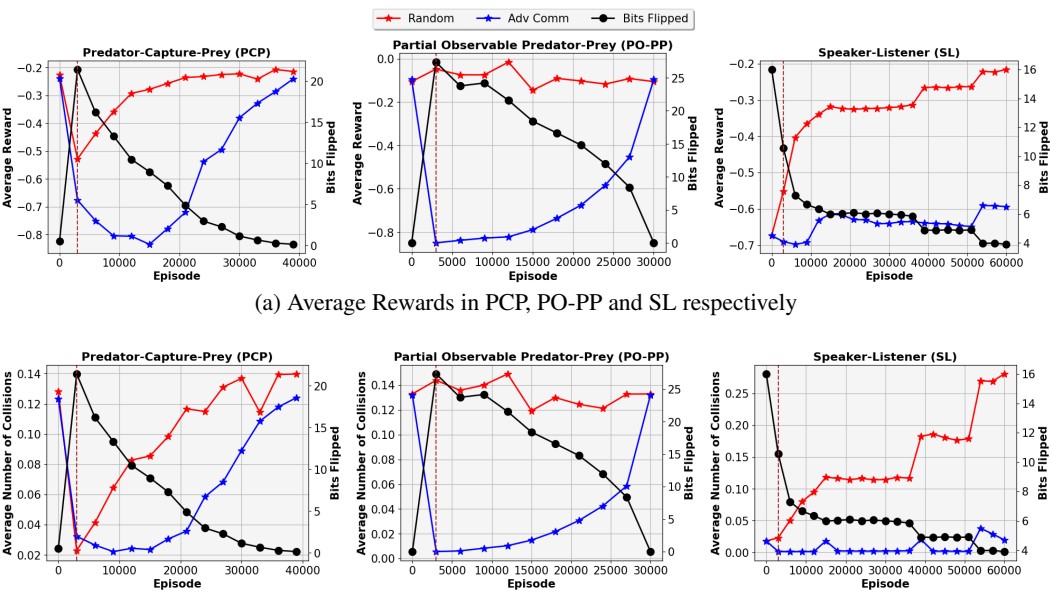

(a) Average Rewards in PCP, PO-PP and SL respectively

(b) Average Collisions for PCP, PO-PP and SL respectively

Figure 7: The average reward and number of collisions during training are displayed for the adversarial policy (blue) and random flipping (red), respectively (using left y-axis). The dotted-brown line represents the episode where the bit-flipping regularization term begins. The regularization term $\mathcal{C}_{flip}$ pushes the adversarial policy to flip fewer bits as training progresses such that adversarial effect becomes weaker. The resulting number of bits flipped is represented as the black curve (using right y-axis). Our adversarial communication policy consistently outperforms the random policy.

## Appendix C   Normalized Score Tables for Attacked Agents

We show detailed tables that quantify the reward and number of collisions for each individual agent here for our adversarial communication with flipping mode, direct mode and the random flipping. Our proposed method is uniformly better than all other strategies across all attacked agents.

Table 10: PCP Reward Scores

|  | Capture Agent 1 | Capture Agent 2 | Average |
|---|---|---|---|
| Adv[Ours] | **0.10±0.06** | **0.12±0.05** | **0.11±0.05** |
| Adv[Direct] | 0.01±0.01 | 0.01±0.01 | 0.01±0.01 |
| Random | 0.01±0.01 | 0.01±0.01 | 0.01±0.01 |

Table 12: SL Reward Scores

|  | Listener |
|---|---|
| Adv[Ours] | **0.13±0.06** |
| Adv[Direct] | 0.11±0.05 |
| Random | 0.04±0.02 |

Table 11: PCP Collision Scores

|  | Capture Agent 1 | Capture Agent 2 | Average |
|---|---|---|---|
| Adv[Ours] | **4.53±2.23** | **4.63±2.23** | **4.58±2.23** |
| Adv[Direct] | 1.07±0.14 | 1.07±0.14 | 1.07±0.14 |
| Random | 0.93±0.29 | 0.99±0.29 | 0.96±0.29 |

Table 13: SL Collision Scores

|  | Listener |
|---|---|
| Adv[Ours] | **31.38±6.51** |
| Adv[Direct] | 30.72±6.19 |
| Random | 13.68±3.19 |

Table 14: PO-PP Reward Scores

|  | PO Agent 1 | PO Agent 2 | PO Agent 3 | PO Agent 4 | Average |
|---|---|---|---|---|---|
| Adv[Ours] | **0.04±0.02** | **0.04±0.02** | **0.04±0.02** | **0.04±0.02** | **0.04±0.02** |
| Adv[Direct] | 0.02±0.01 | 0.02±0.01 | 0.02±0.01 | 0.02±0.00 | 0.02±0.01 |
| Random | 0.00±0.01 | 0.00±0.01 | 0.00±0.01 | 0.01±0.01 | 0.00±0.01 |

Table 15: PO-PP Collision Scores

|  | PO Agent 1 | PO Agent 2 | PO Agent 3 | PO Agent 4 | Average |
|---|---|---|---|---|---|
| Adv[Ours] | **1.41±0.55** | **1.44±0.51** | **1.52±0.51** | **1.42±0.65** | **1.45±0.55** |
| Adv[Direct] | 0.76±0.07 | 0.78±0.07 | 0.82±0.08 | 0.84±0.08 | 0.80±0.07 |
| Random | 0.10±0.87 | 0.34±0.86 | 0.33±0.92 | 0.43±0.88 | 0.30±0.88 |

## Appendix D   Whitebox Analysis

In this section, we compare our methods with a whitebox version of our algorithm (Table 16), where we do not utilize surrogate policies and instead use the true agent policies and reward to learn the adversarial policy. The results show that our adversarial policy achieves comparable collision and reward scores across all domains. The performance between the whitebox method and our method is similar in PCP and PO-PP but has a larger difference in SL. This shows that our surrogate policy can successfully approximate the ground truth agent policies and aid training an adversarial policy.

Table 16: Reward and Collision Normalized Scores

|  | PCP | | PO-PP | | SL | |
|---|---|---|---|---|---|---|
|  | Reward Sc | Collision Sc | Reward Sc | Collision Sc | Reward Sc | Collision Sc |
| Whitebox | 0.12±0.05 | 4.60±2.30 | 0.04±0.02 | 1.37±0.43 | 0.17±0.09 | 41.56±4.98 |
| Adv[Ours] | 0.11±0.05 | 4.58±2.23 | 0.04±0.02 | 1.45±0.55 | 0.13±0.06 | 31.38±6.51 |

# Appendix E   Surrogate Policy Losses

In this section, we show the surrogate policy loss curves. We see that the surrogate policy loss converges relatively quickly before episode 5000. This indicates that while we train for up to 60,000 episodes, much less data could be used to train a stable surrogate policy that can be used for the adversarial communication policy.

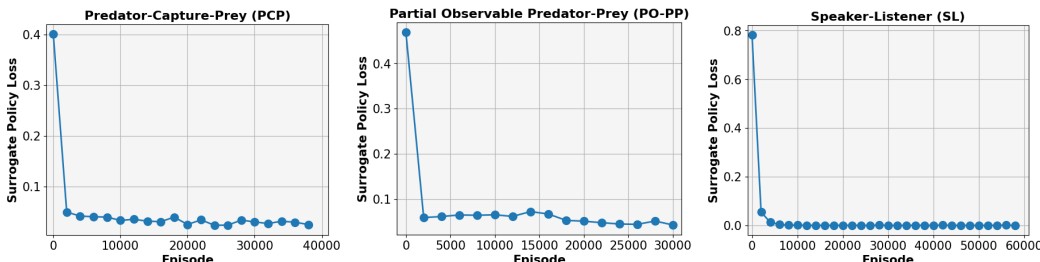

Figure 8: Surrogate Policy Loss in PCP, PO-PP and SL respectively

# Appendix F   Robustness of Analysis Adversarial Policy

We evaluate whether the adversarial policy maintains its performance under minor modifications to the target agent policies. In our experiment, we extend the training of the target team agents by an additional 10,000 episodes, subtly changing their policies. Every 1,000 episodes during this extended training, we gauge the effectiveness of the adversarial attacking policy, where the adversarial policy is frozen and acts without any further training. Our findings indicate that the adversarial policy's performance remains largely consistent, with only a slight decrease in effectiveness and increase in variance (see Figure 9).

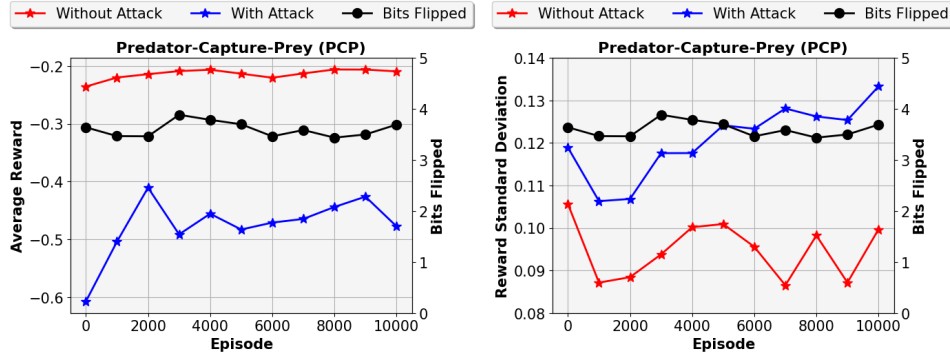

Figure 9: Robustness of Adversarial Policy: Reward Mean (left) and Standard Deviation (right)

