# OpenReview forum: "Hijacking Robot Teams Through Adversarial Communication"
_robot-learning.org/CoRL/2023/Conference — CoRL 2023 Oral_

### Official Review · Reviewer_3YTc · 2023-06-27

**Confidence:** 2
**Originality:** Very Good
**Technical Quality:** Very Good
**Clarity Of Presentation:** Excellent
**Impact:** 4

**Recommendation:**

Strong Accept: I recommend accepting the paper and will argue for my recommendation even if other reviewers hold a different opinion.

**Review:**

Originality: While the idea of hacking/hijacking in a swarm is not new, the specific way that this hijacking is accomplished is (in the opinion of this reviewer) new. The most important difference versus previous work appears to be the limited amount of interaction, e.g., for training purposes, that the black-box adversary model needs with the robot team it is attempting to manipulate.

Quality: The submission is technically sound and the claims well supported by the experimental results. The fact that the method is tested on three different types of multi-robot missions adds to the significance of the work. The paper makes a decent case that the most closely related work is different enough (requiring different training conditions that are non-trivial in the context of an adversarial setting) that comparison should not be required.

Clarity: Is the submission is clearly written, well organized, and easy to understand.

Significance: The results are important in that this shows how the hijacking problem can be addressed in a setting that is arguably new and less strict than that which previous work has addressed.


Relevance: The paper addresses a problem in robot learning, experiments were performed in the robotarium (that is, using real robots).

Limitations: The limitations section exists and is adequate for the purposes of the CoRL conference.


**Quality Of The Limitations Section:**

Additional details required

**Questions For Rebuttal:**


The figure captions and title are way to small... if accepted, can you please make them larger in the final version?

In the figures the use of the word "episode" is somewhat confusing. Is this a training episode?

An additional limitation may be that gathering this much training data (20,000 + training iterations) may be difficult in real life, since there is an implicit assumption that either (a) the adversary has access to a simulation of the team that it can use for learning, or (b) the adversary has access to 20,000 actual interactions in which the team does not detect it is being hacked. Note: this does not detract from the novelty of the work, but it should be mentioned.


**Robotics Focus:**

Sufficient demonstration on hardware

**Summary Of Paper:**

This paper focuses on the problem of tricking a neural network that has been trained for the purposes of multi-agent interaction. The idea is for an adversary to learn how to manipulate the data communicated between robots so as to manipulate the outputs of the model that has been learned by the multi-robot team. The paper looks at such hijacking/hacking in the context of three different multi-agent tasks, and compares the proposed method to randomly manipulating bits in the messages that are communicated between robots.

**Summary Of Recommendation:**

The reviewer has recommended strong accept given that this is the first work that this reviewer is aware of in which a black box hacker is trained to perform a man-in-the-middle attack on a multi-robot learning policy.

---

### Official Review · Reviewer_fyCD · 2023-07-11

**Confidence:** 4
**Originality:** Good
**Technical Quality:** Very Good
**Clarity Of Presentation:** Fair
**Impact:** 2

**Recommendation:**

Strong Accept: I recommend accepting the paper and will argue for my recommendation even if other reviewers hold a different opinion.

**Review:**

Strengths

-  The proposed method improves upon previous works by not requiring interaction with the environment, thus, increasing its potential to not be detectable.
-  The proposed method is tested on hardware, using the Robotarium multi-robot test bed.

Weaknesses

-  I would elaborate on how the attacker can have access to the observations of the agents.  If these observations are, for instance, camera images, how the attacker can get access to these images or to a processed form of those?

-  I would discuss why the proposed attack scheme is not detectable.  For example, why it cannot be the case that the altered messages don't make sense? In such a case ---altered message doesn't make sense--- the attack is immediately detected.   This can happen when, for instance, the altered messages prescribe locations that are outside the boundary of operation.

**Quality Of The Limitations Section:**

Limitations are not well addressed

**Questions For Rebuttal:**

Please see all my comments in the weakness section above.  I would also elaborate on the robustness of the learned attack policy.  How effective a learned attack policy remain when the robots alter their message policy?

**Robotics Focus:**

Sufficient demonstration on hardware

**Summary Of Paper:**

The paper proposes a communication attack method to compromise multi-robot coordination.  The method requires knowledge of the robots' observations as well as of their message policies given the observations. The proposed method is tested on the Robotarium platform in target tracking tasks.

**Summary Of Recommendation:**

The paper proposes an interesting method for compromising multi-robot direction.  In my opinion, the current version of the paper would benefit from clarifying the practicality of the assumptions and the degree of detectability of the designed attacks.

UPDATE: I update my score to strong accept since the authors addressed all my during with their rebuttal.

---

### Official Review · Reviewer_ooVv · 2023-07-19

**Confidence:** 3
**Originality:** Good
**Technical Quality:** Very Good
**Clarity Of Presentation:** Very Good
**Impact:** 4

**Recommendation:**

Strong Accept: I recommend accepting the paper and will argue for my recommendation even if other reviewers hold a different opinion.

**Review:**

Originality) Even though the idea of an attack on communication in the MARL system is not original [36, Mis-spoke or mislead: Achieving Robustness in Multi-Agent Communicative Reinforcement Learning], the authors increase the applicability of such an attack by introducing a successful black-box algorithm.

Quality ) The theoretical background of the paper is convincing, however, the experimental part raises many questions for me. At 219 the authors state that collision numbers increase when we decrease the number of bits flips, which sounds counterintuitive. It is not clear to me the meaning of the episode axis in Figure. 3: If it illustrates the training process of malicious policy then why does it perform better than random flip at point zero? In Table 1 the best algorithm has the most number of collisions, however, in Table 2, it is the opposite.

Clarity ) The authors do not provide any details on the learning process i.e. hyper parameters, number of bits in communication messages, enough number of observations to successfully perform such attack. It will be hard to reproduce the results.

Significance) I think the results of this paper could be useful because they propose a real-world applicable attack on MARL systems and researchers should be aware of such algorithms, thinking about effective defenses.

Relevance) I think it is a relevant paper because the authors present an algorithm that could compromise real-world systems. In addition, they demonstrate the effectiveness of their approach on physical swarm robots.

Limitations) The authors do not say about the time requirement for their agent to perform such an attack in terms of observed interactions or real-time, it is important because a man-in-the-middle attack will be detected if it requires a lot of time. Otherwise, limitations are addressed well.

**Quality Of The Limitations Section:**

Limitations are addressed clearly

**Questions For Rebuttal:**

* Explain or better represent the results of experiments, as was stated in the review

* Add information about hyperparameters, and the amount of data to train adversarial policy to encourage reproducibility

* On lines 211 and 214 presented formulas without specific values of hyperparameters, variables beta, and epsilon are not explained at all

* Include the amount of acquired training data to the limitations of your approach

**Robotics Focus:**

Sufficient demonstration on hardware

**Summary Of Paper:**

The authors of the paper present their approach to a black-box attack multi-agent reinforcement learning system. They draw attention to the importance of secure communication between agents during solving tasks and create their malicious model to alternate these communication messages. In opposite to other articles in this field, authors do not interact with the environment while training their malicious policy; they neither have access to ground truth agents’ policies or rewards. The authors consider binary communication channels and compare their approach with the random bit-flipping policy. The proposed method shows better metrics than baseline on three different Dec-POMDPP environments in simulation and Robotarium physical robots.

**Summary Of Recommendation:**

It is an important paper that could improve the safeness of MARL real-world systems, providing an effective black-box attack on such systems without interaction with the environment or access to the reward. However, the experimental section is unclear and, in my opinion, requires more explanations and information.

After author response: I appreciate the author's efforts to improve the clarity of the paper. Also, all of my concerns were well-addressed. Considering this I am upgrading my score to strong accept.

---

### Official Review · Reviewer_N9UF · 2023-07-21

**Confidence:** 4
**Originality:** Very Good
**Technical Quality:** Good
**Clarity Of Presentation:** Good
**Impact:** 4

**Recommendation:**

Strong Accept: I recommend accepting the paper and will argue for my recommendation even if other reviewers hold a different opinion.

**Review:**

# Strengths

## Originality/Significance

The paper considers a novel setting for adversarial attacks that is relevant and interesting. The attack setting is plausible for several applications and there is a good potential for follow-on work. The setting, where attackers can not experiment directly on victim policies, is challenging and potentially valuable.

## Quality

The approaches in the paper are sound and the evaluation results are good.

# Weaknesses

## Clarity

There are a few places where the clarity can be improved:

* Can you make the inputs to Algorithm 1 more clear? I realize they are defined elsewhere, but it would help to be redundant here, IMO
* Figure 3: can you add more to the caption to help readers interpret this figure? I had a hard time figuring out what is going on (still not sure I got it tbh...)
* Can you clarify in 5.4 if this is in simulation or on a physical system? This was hard for me to figure out.

## Quality

I also have some questions about the evaluation methodology, please see that portion of the review.

**Quality Of The Limitations Section:**

Additional details required

**Questions For Rebuttal:**

## Evaluation Baselines

* Can you justify your choice of baselines? Why did you pick the bit-flipping method? Why did you control the bit flips in the way you did? Are there alternative random baselines you considered?
* Can you compare against the setting without surrogate policies? How much harder is the setting where you have to learn surrogate policies for victims?

## Discussion questions

* It seems like this attack works to prevent a system from accomplishing its goal. If, instead, you had an, alternative, desired goal for the system, could you modify the attack to steer the system towards arbitrary goals?

**Robotics Focus:**

Highly relevant to robotics but no hardware experiments

**Summary Of Paper:**

The paper proposes a novel security context for adversarial attacks on multi-agent systems. The proposal is that, through some mechanism, an adversary is able to perform a man-in-the-middle attack on a multi-agent robot team. This allows the attacker to modify communication signals and influence behavior. They consider a setting where the attacker acts in a one-shot fashion to execute the attack but observes the victim's behavior beforehand.

**Summary Of Recommendation:**

I'm fairly positive about this paper and I think it studies a problem that is quite interesting. I think the approach is sound and has some promise. I have a few reservations about the evaluation baselines and thoroughness which hold me back from a strong accept. If the authors can address those in the rebuttal I am likely to increase my score.

Update after rebuttal: I am satisfied with the response and have updated my score accordingly. I recommend that the authors update the paper to make sure that the justification of the baselines is clear to readers. Additionally, I think that a discussion of driving the system to arbitrary goals may help open up interesting avenues for future work.

---

### Decision · Program_Chairs · 2023-08-30

**Decision:**

Accept (Oral)

**Comment:**

The paper considers the case of a multi-agent scenario where the agents communicate with each other. An adversarial policy is used for tricking the agents to perform actions which are not originally desired. The method is evaluated in simulated benchmark tasks and using physical swarm robots in the Robotarium. The paper studies an interesting problem setting and the proposed approach gets high performance in the experiments.